# Effects of Age, Metabolic and Socioeconomic Factors on Cardiovascular Risk among Saudi Women: A Subgroup Analysis from the Heart Health Promotion Study

**DOI:** 10.3390/medicina59030623

**Published:** 2023-03-21

**Authors:** Hayfaa Wahabi, Samia Esmaeil, Rasmieh Zeidan, Amel Fayed

**Affiliations:** 1Research Chair for Evidence-Based Health Care and Knowledge Translation, King Saud University, P.O. Box 800, Riyadh 11421, Saudi Arabia; 2Department of Family and Community Medicine, College of Medicine, King Saud University Medical City, P.O. Box 800, Riyadh 11421, Saudi Arabia; 3Cardiac Sciences Department, College of Medicine, King Saud University, P.O. Box 800, Riyadh 11421, Saudi Arabia; 4Clinical Sciences Department, College of Medicine, Princess Nourah bint Abdulrahman University, P.O. Box 84428, Riyadh 11671, Saudi Arabia

**Keywords:** Saudi women, cardiovascular disease, risk factors, menopause, pregnancy

## Abstract

*Background*: Cardiovascular disease (CVD) remains the leading cause of death in women. Along with the effect of age on the risk of CVD, the reproductive profile of women can influence cardiac health among women. *Objectives*: The objective of this study is to investigate the influence of age and reproductive stages on the development and progression of cardiovascular disease risks in Saudi women. *Methods*: For this study, we included 1907 Saudi women from the Heart Health Promotion Study. The study cohort was divided into five age groups (less than 40 years, 40–45 years, 46–50 years, 51–55 years, and ≥56 years). The cohort stratification was meant to correspond to the social and hormonal changes in women’s life, including reproductive, perimenopausal, menopausal, and postmenopausal age groups. The groups were compared with respect to the prevalence of metabolic, socioeconomic, and cardiac risks, and the age group of less than 40 years was considered as the reference group. The World Health Organization stepwise approach to chronic disease risk factor Surveillance-Instrument v2.1 was used in this study to collect the anthropometric and biochemical measurements and the Framingham Coronary Heart Risk Score was used to calculate the cardiovascular risk (CVR). Logistic regression analysis was conducted to assess the independent effect of age on CVD risks after adjustment of sociodemographic factors. *Results*: Metabolic and CVR increased progressively with the increase in age. There was a sharp increase in obesity, hypertension, diabetes, and metabolic syndrome, from the age group <40 years to 41–45 years and then again between the age groups of 46–50 and ≥56 years. A similar noticeable increase in metabolic risk factors (high cholesterol, high triglyceride, high Low-Density Lipoprotein) was observed between the age group <40 years and 41–45 years, but with a steady increase with the increase in age between the other age groups. The high and intermediate Framingham Coronary Heart Risk Scores showed a progressive increase in prevalence with the increase in age, where the proportion doubled from 9.4% at the age group 46–50 years, to 22% at the age group 51–55 years. It doubled again at the age group ≥56 years to 53%—these sharp inflections in the risk of CVD correspond to the women’s reproductive lives. *Conclusions*: In Saudi women, CVR increases with the increase of age. The influence of pregnancy and menopause is apparent in the prevalence of increased risks for cardiovascular and metabolic diseases.

## 1. Introduction

Globally, 35% of death in women is due to cardiovascular disease (CVD), with an estimated 8.9 million deaths in 2019 [1]. The Middle East, including Saudi Arabia, is among the regions with the highest mortality in women from CVD with 486 deaths per 100,000, which is considerably high compared to other high-income countries such as Australia and North America with a reported <130 deaths per 100,000 [2]. Ischemic heart disease is leading cause of death in women from CVD worldwide including in Saudi Arabia, with stroke coming in second place. 

The main risk factor for CVD in women in the Middle East and North Africa is hypertension, which imposes higher risk of myocardial infarction in women than in men, followed by dyslipidemia with total cholesterol showing considerable increase following the menopause [2,3,4]. Other equally important and interrelated risk factor for CVD in women are high body mass index (BMI), sedentary lifestyle and diabetes. It is documented that older women lead more sedentary lifestyles than older men [5]. Physical inactivity is associated with obesity, diabetes, and hypertension. Observation showed that a similar increase in both genders in BMI is associated with a greater increase in systolic blood pressure in women than in men [6]. Furthermore, the risk of CVD attributed to obesity is 20% more in women compared to the risk in men of the same age group [7].

Most of the studies which investigated CVD and risk factors in women in Saudi Arabia were either small cross-sectional studies targeting young college students [8,9,10], mostly investigating one risk factor for CVD [11,12], or studies which included both men and women but did not include analysis to investigate the influence of age, or the hormonal changes of pregnancy and the menopause on the CVD risk factors [13,14,15,16]. 

The objective of this study is to investigate the influence of age on the development and progression of CVD and cardiovascular risk (CVR) in women which will define the target age group for intervention to reduce mortality and morbidity for CVD among women in Saudi Arabia. 

## 2. Materials and Methods

### 2.1. Consent and Ethics

The study followed the standards of the Helsinki Declaration after receiving approval from King Saud University’s Institutional Review Board (IRB) (reference number 13–3721). We invited 5200 individuals to participate in the study and 4500 participants agreed to be enrolled (response rate of 87%). All participants signed informed consent forms.

### 2.2. Study Setting

The original cohort included 4500 participants recruited from employee clinics in King Saud University Hospital that serve the employees and their families. The data collection extended for a period of 9 months (from 8 July 2013 to 30 April 2014) and the first report was published in 2016.

### 2.3. Study Population and Sampling Technique

For this study we included only 1907 Saudi women from the total cohort. We excluded pregnant women from the study. Considering prevalence of obesity as of 25% ± 5% (*p* < 0.01), a power of >0.9 was calculated using STATA/IC14.2. 

The study cohort was divided into five age groups (less than 40 years, 40–45 years, 46–50 years, 51–55 years and ≥56 years). The stratification of the cohort was meant to approximately correspond to the social and hormonal changes in women’s life including reproductive, perimenopausal, menopausal and postmenopausal age groups. The groups were compared with respect to the prevalence of the metabolic and the socioeconomic CVD and CVR. 

### 2.4. Data Collection and Physical Measurements

The sociodemographic data (age, marital status, occupation, and educational attainment), data about tobacco use, physical activity, healthy diet, and anthropometric and biochemical measurements were collected using the World Health Organization (WHO) stepwise approach to chronic disease risk factor Surveillance-Instrument v2.1 [17]. 

All participants were required to fast for at least 12 h before giving blood samples. Glycosylated hemoglobin (HbA1c), high-density lipoprotein cholesterol (HDL-C), low-density lipoprotein cholesterol (LDL-C), total cholesterol (TC), and triglycerides (TG) were measured. 

### 2.5. Study Variables


Obesity: Weight and height were measured for all participants. Weight was measured to the nearest 10 g, while height was measured to the nearest 0.1 cm.
⮚The Body Mass Index (BMI): was calculated using the formula BMI = weight (kg)/height (m^2^). Based on the BMI, the study population was divided into five groups: underweight, normal weight, overweight, obese and morbidly obese, (<18.5; 18.5–24.9; 25–29.9; 30–34.9; ≥35 kg/m^2^), respectively [18].⮚Central Obesity: The waist circumference (WC) was measured in centimeters to the nearest 0.1 cm, using a flexible non-stretchable plastic tape, in a standing relaxed position, during expiration, at the midline between the lower costal margins and the iliac crest parallel to the floor. A WC of 88 cm was used for diagnosis of central obesity among women, which are cut-off values reported to be applicable to Arab ethnicities [19].Current smokers: were classified as individuals who had smoked at least one cigarette per day for the previous six months, one cigar or water pipe weekly for the last six months, or one waterpipe tobacco smoke/shisha session each month for the prior three months [20].Physical inactivity: participants were deemed physically inactive if they did not meet any of the following WHO standards: 150 min of moderate activity each week, or 60 min of vigorous activity [21].Low fruit and vegetable intake: According to the WHO, any subject who had less than five servings (400 gm) of fruit and/or vegetables per day was considered as having inadequate intake [22].Hypertension: Both systolic and diastolic pressures were measured, at two readings, set five minutes apart; the average of the two readings was used. Hypertension was defined as being previously diagnosed as hypertensive and currently using any anti-hypertensive medications or having high blood pressure readings according to Seventh Report of the Joint National Committee on Prevention, Detection, Evaluation, and Treatment of High Blood Pressure (JNC7) [23].Diabetes mellitus was defined as per WHO and American Diabetes Association criteria, or by subject reporting of being previously diagnosed as diabetic and using anti-diabetes medication [24].Cardiovascular risk (CVR) scores were calculated for all participants using Framingham Coronary Heart Risk Score (FRS) which is one of the most extensively used cardiovascular risk calculators in clinical practice. It was used to calculate the 10-year risk of coronary heart disease where the cohort was sub-divided according to their scores into three categories: low risk score (<10%), intermediate (10–20%), and high (>20%) [25].Metabolic Syndrome (MetS): If participants satisfied at least three of the five criteria listed in the Third Report of the National Cholesterol Education Program (NCEP) Adult Treatment Panel III) (NCEP-ATPIII) criteria, they were considered to have metabolic syndrome [26].Dyslipidemia: dyslipidemia was considered according to definitions adopted by the National Cholesterol Education Program (NCEP) criteria for dyslipidemia (elevated cholesterol, elevated TG, high HDL-C level and low LDL-C) [26].


### 2.6. Statistical Analysis

Continuous variables, interval and ratio variables were reported as means with standard deviations. Categorical variables were presented as frequencies with equivalent percentages, and Pearson’s chi-square test was used for comparison of different proportions. Logistic regression analysis was conducted to assess the independent effect of age on CVR after adjustment of sociodemographic factors (education, occupation, and marital status) and considered the younger age group (<40 years) as the reference group. When assessing the CVR using the FRS, we aggregated the intermediate and high-risk groups as one category to convert the FRS into binary variable. Predictive probability (PP) of outcomes according to different age groups was plotted with its 95% confidence intervals (CI). Statistical analyses of the data were done using SPSS V.26.0 statistical package (IBM SPSS) and STATA version 16. 

## 3. Results

A total of 1907 women were included in this study. The socioeconomic characteristics with impact on cardiovascular risks are shown in Table 1. While 46.5% of the age group <40 years were single, only 4.6% of the age group 40–45% were in this category. The range of the prevalence of unhealthy dietary habits (84–92%), and physical inactivity (83–93%) was high across all age groups, despite the significant difference in prevalence of employment between the <40 years old and the older groups (Table 1). However, the prevalence of tobacco smoking was low across all age groups (Table 1). 

Cardiometabolic risks increased progressively with the increase in the age of the cohort (Table 2 and Figure 1). In addition, the analysis showed a marked increase in HTN, DM, and MetS, between the age group <40 years and 41–45 years and between the age groups of 46–50 and ≥56 years (Table 1). Similar noticeable increase in overweight/obesity, and dyslipidemias was observed between the age group <40 years and 41–45 years, but with steady increase with the increase in age between the other age groups (Table 2). The intermediate and high FRS showed a progressive increase in prevalence with the increase in age. There are two points of sharp increase in the proportion of women with intermediate and high scores, where the proportion doubled from 9.4% at the age group 46–50 years, to 22% at the age group 51–55 years, and it doubled again at the age group ≥56 years to 53% (Table 2 and Figure 1). 

Predicted probability of high and intermediate CVR as measured by FRS was derived from the regression models after adjustment of socioeconomic factors. There was an escalating trend of the probabilities of CVR across the age groups that reached above 50% (95% C.I. = 47–59%) among women aged 56 years (Figure 2).

## 4. Discussion

The results of this study showed that there was a progressive increase in the risks and probability of developing CVD with the increase of age in Saudi women, and that there was a high proportion of women in this cohort who have unhealthy food habits and who had led a sedentary lifestyle. Furthermore, the prevalence of risk factors showed a sharp and progressive increase after the age of 40–45 years with noticeable association of transition from single to married social status and with the completion of reproductive stage of the women’s life. 

Similar to our results, previous reports confirmed that ageing is one of the main and independent risk factors for CVD both in males and females [27]. This observation is mainly due to the changes at the cellular level of the heart and the vasculature with increased accumulation of collagen and depletion of elastin [27]. These changes lead to deterioration in the function and morphology of the heart and the vascular system; including high systolic blood pressure, widening of the pulse pressure together with atrial dilatation and ventricular hypertrophy [28] due to stiffness of the aorta and generalized endothelial dysfunction and central arterial stiffness of the vascular system. 

The clinical manifestations of the deteriorating function of the cardiovascular system include; hypertension, atrial fibrillation [29], heart failure [30], stroke and ischemic heart disease [31]. Often women develop non-coronary obstructive heart diseases. One example is Takotsubo Syndrome that seems to afflict the female sex almost exclusively [32]. The pivotal effect of aging as risk factor for CVD is clearly shown in Figure 1 and Figure 2, which showed a steady increment of prevalence of CVR scores with age and substantial increase in the probability of developing CVD with the increase in age. This observation may be explained by the known effects of advanced age equally on the other risk factors including hypertension, as shown above, glucose metabolism, and obesity. Recent studies provided evidence on the dysregulation of glucose metabolism in elder men and women with defective insulin secretion, high fasting blood glucose, delayed postprandial glucose clearance and increased liver production of glucose compared to young individuals [33]. Although old women have slightly different defects of glucose regulation compared to old men, both genders are at increased risk of developing diabetes as they get older [33]. 

Obesity, especially visceral, is a proven risk factor for insulin resistance, diabetes, dyslipidemia and CVD [34]. In this study the prevalence of obesity, as indicated by BMI, is very high, but it plateaus after the age of 45 years as shown in Figure 1, unlike the prevalence of central obesity which showed an incremental course with the advancing age of the women; hence, it corresponds to the increased prevalence of the risk of CVD. Previous studies confirmed our findings that as women get older, they are more borne to develop visceral obesity, which is more detrimental as a risk factor for CVD than total body fat [35,36,37]. Such change in the distribution of body fat is mediated by estrogen deficiency during the perimenopause and menopause stage of the woman’s reproductive life [38,39]. 

Visceral obesity is the main source of the high levels of LDL-C, free fatty acids, and insulin resistance observed with the advance of the individual’s age [40], which has been confirmed by the findings of this study (Table 2). 

Other events in women reproductive life have been linked to cardiometabolic risks for CVD including gestational diabetes, prepregnancy obesity, postpartum weight retention and preeclampsia [41,42,43,44]. Almost all of these conditions are quite prevalent among Saudi women during pregnancy and the postpartum period, as proven by the largest study conducted in Riyadh city which included more than 14,000 women and their neonates [45], which may indicate that Saudi women are at greater risk of developing CVD compared to women in other high income countries. In this study there is a noticeable increment in all CVD risk factors after the age of 40 years (Figure 1 and Figure 2 and Table 2). This age group corresponds to a social status of married women who have completed their families and approaching the perimenopausal period of their reproductive life, which indicate not only the effect of age on CVR, but the detrimental effect of postpartum weight retention, events during pregnancy such pre-eclampsia, gestational diabetes, and estrogen deficiency.

We are aware of the limitations of this study including the lack of data and analysis of the influence of reproductive life events on the risk of developing cardiovascular disease including the number of pregnancies and the occurrence of events such as gestational diabetes and pre-eclampsia. However, the completion of all records in this large size cohort revealed robust evidence for other important variables. We should also refer to the use of FRS instead of using SCORE2 & SCORE2-OP [46], as we were not able to define studies conducted among Saudis that used this score for CVR assessment; however, we are planning to re-analyze our database and compare the results of these scores with FRS in future reports.

## 5. Conclusions

In Saudi women, CVR increases with the increase of age. The influence of pregnancy and menopause are apparent in the prevalence of increased risks for cardiovascular and metabolic diseases.

## Figures and Tables

**Figure 1 medicina-59-00623-f001:**
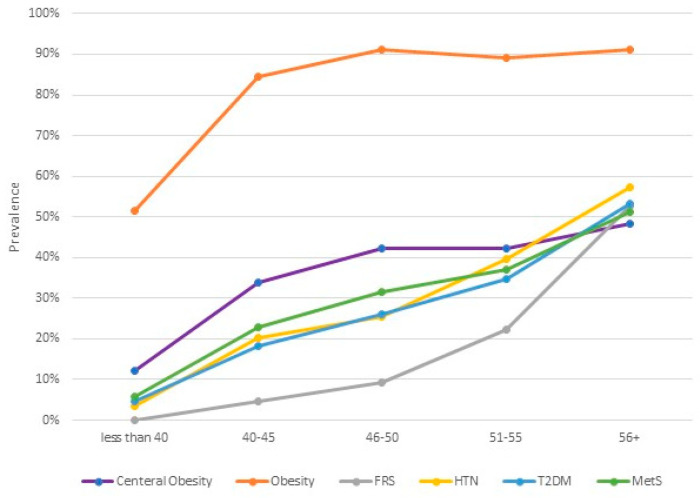
Prevalence of different cardiometabolic risks among the studied sample according to their age. Obesity: Body Mass index more than 25 kg/m^2^, FRS: High/intermediate Framingham Coronary Heart Risk Score, HTN: Hypertension, T2DM: Type 2 Diabetes Mellites, MetS: Metabolic Syndrome Score.

**Figure 2 medicina-59-00623-f002:**
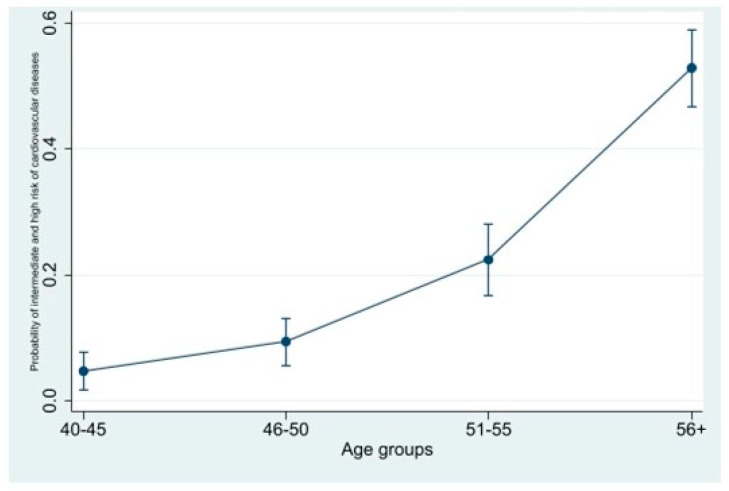
Predictive probabilities of intermediate/high cardiovascular risk among different age groups.

**Table 1 medicina-59-00623-t001:** Characteristics of the studied sample according to different age groups.

		Women Age										
		Less than 40		40–45		46–50		51–55		56+		*p*-Value
		No	%	No	%	No	%	No	%	No	%	
**Marital Status**	Married	505	(49.5)	169	(88.0)	226	(96.2)	197	(93.8)	208	(83.2)	<0.01
	Single	474	(46.5)	9	(4.7)	2	(0.9)	0	(0.0)	4	(1.6)	
	Widowed and divorced	41	(4.0)	14	(7.3)	7	(3.0)	13	(6.2)	38	(15.2)	
**Education**	College and above	648	(63.5)	116	(60.4)	111	(47.2)	89	(42.4)	73	(29.2)	<0.01
	School (up to 12 years)	362	(35.5)	69	(35.9)	105	(44.7)	98	(46.7)	84	(33.6)	
	Illiterate	10	(1.0)	7	(3.6)	19	(8.1)	23	(11.0)	93	(37.2)	
**Occupation**	Student	201	(19.7)	2	(1.0)	2	(0.9)	3	(1.4)	0	(0.0)	<0.01
	Employee	385	(37.7)	53	(27.6)	44	(18.7)	32	(15.2)	33	(13.2)	
	Housewife	434	(42.5)	137	(71.4)	189	(80.4)	175	(83.3)	217	(86.8)	
**Smoking**	Smoker	27	(2.6)	7	(3.6)	5	(2.1)	6	(2.9)	5	(2.0)	0.83
**Dietary Habits**	Inadequate fruits/vegetables	940	(92.2)	164	(85.4)	199	(84.7)	183	(87.1)	212	(84.8)	<0.01
**Physical activity**	Physically inactive	858	(84.1)	167	(87.0)	220	(93.6)	193	(91.9)	234	(93.6)	<0.01

Data are presented as frequency and (%), women age in years, chi-square test was used for testing the association between each variable and age category.

**Table 2 medicina-59-00623-t002:** Cardiometabolic profile of the studied sample among different age groups.

		Women Age										*p*-Value
		Less than 40		40–45		46–50		51–55		56+		
		No	%	No	%	No	%	No	No	No	%	
**Obesity**	Underweight < 18.5	56	(5.5)	1	(0.5)	0	(0.0)	2	(1.0)	1	(0.4)	<0.01
	Normal weight 18.5–24.9	440	(43.1)	29	(15.1)	21	(8.9)	21	(10.0)	21	(8.4)	
	Overweight 25–29.9	290	(28.4)	55	(28.6)	79	(33.6)	69	(32.9)	75	(30.0)	
	Obese 30–34.9	146	(14.3)	53	(27.6)	62	(26.4)	66	(31.4)	89	(35.6)	
	Morbid obese ≥ 35	88	(8.6)	54	(28.1)	73	(31.1)	52	(24.8)	64	(25.6)	
	Overweight and obesity	524	(51.3)	162	(84.3)	214	(91,1)	187	(89.9)	228	(91.2)	
	Central Obesity	125	(12.3)	65	(33.9)	99	(42.2)	89	(42.4)	121	(48.4)	
**Hypertension**	Normal	984	(96.5)	153	(79.7)	175	(74.5)	127	(60.5)	107	(42.8)	<0.01
	HTN	36	(3.5)	39	(20.3)	60	(25.5)	83	(39.5)	143	(57.2)	
**Diabetes**	Normal	973	(95.4)	157	(81.8)	174	(74.0)	137	(65.2)	117	(46.8)	<0.01
	T2DM	47	(4.6)	35	(18.2)	61	(26.0)	73	(34.8)	133	(53.2)	
**Dyslipidemias**												
Triglycerides	normal	952	(93.3)	155	(80.7)	178	(75.7)	159	(75.7)	193	(77.2)	<0.01
	High level of TG	68	(6.7)	37	(19.3)	57	(24.3)	51	(24.3)	57	(22.8)	
Total Cholesterol	Normal	729	(71.5)	100	(52.1)	120	(51.1)	105	(50.0)	140	(56.0)	<0.01
	High level of TC	291	(28.5)	92	(47.9)	115	(48.9)	105	(50.0)	110	(44.0)	
low-density lipoprotein	Normal	779	(76.4)	109	(56.8)	138	(58.7)	122	(58.1)	156	(62.4)	<0.01
	High level of LDL-C	241	(23.6)	83	(43.2)	97	(41.3)	88	(41.9)	94	(37.6)	
High-density lipoprotein	Normal	894	(87.6)	156	(81.3)	192	(81.7)	175	(83.3)	205	(82.0)	0.02
	Low level of HDL-C	126	(12.4)	36	(18.8)	43	(18.3)	35	(16.7)	45	(18.0)	
**FRS**	Low risk < 10%	1020	(100.0)	183	(95.3)	213	(90.6)	163	(77.6)	118	(47.2)	<0.01
	Intermediate risk 10–20%	0	(0.0)	8	(4.2)	22	(9.4)	42	(20.0)	82	(32.8)	
	high risk > 20%	0	(0.0)	1	(0.5)	0	(0.0)	5	(2.4)	50	(20.0)	
**Metabolic Syndrome**	Normal	960	(94.1)	148	(77.1)	161	(68.5)	132	(62.9)	122	(48.8)	<0.01
	MetS	60	(5.9)	44	(22.9)	74	(31.5)	78	(37.1)	128	(51.2)	

Data are presented as frequency and (%), women age in years, HTN: Hypertension, T2DM: Type 2 Diabetes Mellites, TG: Triglycerides, TC: Total cholesterol, LDL-C: Low-density lipoprotein, HDL: High-density lipoprotein, FRS: Framingham Coronary Heart Risk Score, MetS: Metabolic syndrome.

## Data Availability

Most of the data needed is included in the published article. However, more data is available from the King Saud University Ethics Committee for researchers who meet the criteria for access to confidential data. The ethics committee contact details for data requests are: irb@ksu.edu.sa. This contact point is completely independent of all researchers.

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
