# Peer review of "Effects of Age, Metabolic and Socioeconomic Factors on Cardiovascular Risk among Saudi Women: A Subgroup Analysis from the Heart Health Promotion Study"

_medicina, 2023, doi:10.3390/medicina59030623_

Round 1

Reviewer 1 Report

I read with great interest the paper authored by Hayfaa Wahabi et al. The study highlights the important topic of gender medicine and, in particular, analyses the cardiovascular risk among Saudi women. The study take into account not only the classical risk factors, but also the socio-economic factors and I really trust that it is a keystone in order to better understand the gender differences in cardiovascular disease.

However, in my humble opinion, the study need some major and minor revision before the publication:

1.       Please, use the template of the journal;

2.       If not strictly necessary, remove brackets and abbreviations from the title;

3.       Remove the yellow highlight on the first author;

4.       In the abstract, insert the background;

5.       In material an methods, please specify what is the temporal period of the research and what is the temporal period of revealments;

6.       In the section “study population and sampling technique”, please correct the number of group in “five age groups” according with the explanation in the subsequent round brackets;

7.       At the end of page 3, please extend the acronym “JNC7”;

8.       In the study, cardiovascular risk scores were calculated using Framingham Coronary Heart Risk Score (European Respiratory Journal. 2010; 36(5):1002-6). In my opinion, in order to estimate 10-years risk of (fatal or non-fatal) CV events, could be more appropriate and updated to use “SCORE2 & SCORE2-OP” as recommended by the guidelines of European Society of Cardiology released in 2021 (European Heart Journal. 2021; 42:3227-3337 and free available at https://www.escardio.org/Guidelines/Clinical-Practice-Guidelines/2021-ESC-Guidelines-on-cardiovascular-disease-prevention-in-clinical-practice). I would like to suggest, if possible, compatibly with data set and availability of the resources, to change it.

9.       In Table 1, Table 2, Figure 1 and Figure 2 the data of the fifth group (population > 56 years old) are missing. Please, change the layout of the figures in order to view them in their entirety.

10.   In Figure 1, please moves the legend beyond the caption.

11.   In the discussion, at page 9 and after the reference (30),  I would like to suggest to insert the following sentence: “Often the women develop non-coronary obstructive heart diseases. One example is the tako-tsubo syndrome that seems to afflict the female sex almost exclusively.” and cite “Pulmonary embolism in a patient with apical ballooning syndrome. Fedele F, Gatto MC. J Cardiovasc Med (Hagerstown). 2012 Jan;13(1):56-9.”

12.   At page 9 change “&” with “and” (e.g. Figure 1 and Figure 2) (e.g. Figure 1, Figure 2 and Table 1).

13.   Please indicate in the text Figures and Tables with the first letter in capital letters (e.g. Figure 1, Table 1 etc.)

In conclusion I would warmly recommend to the authors performing the revisions required in order to complete and publish the very well study proposed.

Author Response

We would like to thank our reviewer for his valuable constructive comments and appreciate his detailed and comprehensive review.

Kindly find our point-by-point reply:

  1. Please, use the template of the journal.

Reply: The tables and figures were corrected to fit in the PDF format and are clearly visible now.

  1. If not strictly necessary, remove brackets and abbreviations from the title.

Reply: Done

  1. Remove the yellow highlight on the first author.

Reply: Done

  1. In the abstract, insert the background.

Reply: Done

 “Background: Cardiovascular disease (CVD) remains the leading cause of death in women. Along with the effect of age on the risk of CVD, the reproductive profile of women can influence the cardiac health among women.”

  1. In material and methods, please specify what is the temporal period of the research and what is the temporal period of revealments.

Reply: Done

“The data collection extended for a period of 9 months (from 8 July 2013 to 30 April, 2014) and the first report was published in 2016.”

  1. In the section “study population and sampling technique”, please correct the number of group in “five age groups” according with the explanation in the subsequent round brackets;

Reply: Done

  1. At the end of page 3, please extend the acronym “JNC7”;

Reply: Done

  1. In the study, cardiovascular risk scores were calculated using Framingham Coronary Heart Risk Score (European Respiratory Journal. 2010; 36(5):1002-6). In my opinion, in order to estimate 10-years risk of (fatal or non-fatal) CV events, could be more appropriate and updated to use “SCORE2 & SCORE2-OP” as recommended by the guidelines of European Society of Cardiology released in 2021 (European Heart Journal. 2021; 42:3227-3337 and free available at https://www.escardio.org/Guidelines/Clinical-Practice-Guidelines/2021-ESC-Guidelines-on-cardiovascular-disease-prevention-in-clinical-practice). I would like to suggest, if possible, compatibly with data set and availability of the resources, to change it.

Reply: We definitely appreciate this very valuable suggestion as it can shed light on some very new prospects of the CVR among our studied population. However, it will need to change all the results of the current report and will require re-analysis and interpretation of the results. It will also need further clarifications from SCORE2& SCORE2-OP about the geographical consideration and validation of its use among the Saudi population.

  1. In Table 1, Table 2, Figure 1 and Figure 2 the data of the fifth group (population > 56 years old) are missing. Please, change the layout of the figures in order to view them in their entirety.

Reply: Done

  1. In Figure 1, please moves the legend beyond the caption.

Reply: Done

  1. In the discussion, at page 9 and after the reference (30),  I would like to suggest to insert the following sentence: “Often the women develop non-coronary obstructive heart diseases. One example is the tako-tsubo syndrome that seems to afflict the female sex almost exclusively.” and cite “Pulmonary embolism in a patient with apical ballooning syndrome. Fedele F, Gatto MC. J Cardiovasc Med (Hagerstown). 2012 Jan;13(1):56-9.”

Reply: the paragraph was added along with its reference.

  1. At page 9 change “&” with “and” (e.g. Figure 1 and Figure 2) (e.g. Figure 1, Figure 2 and Table 1).

Reply: Done

  1. Please indicate in the text Figures and Tables with the first letter in capital letters (e.g. Figure 1, Table 1 etc.)

Reply: Done

In conclusion I would warmly recommend to the authors performing the revisions required in order to complete and publish the very well study proposed.

Reply: Thank you so much for these informative comments

Reviewer 2 Report

I strongly suggest that authors learn the published article on this topic, Int. J. Environ. Res. Public Health 2022, 19(22), which is more comprehensive than the current manuscript.  This content of current manuscript is not enough.

1. What is the main question addressed by the research? A subgroup analysis was conducted by authors to Effects of age, metabolic and socioeconomic factors on cardiovascular risk among Saudi women.
2. Do you consider the topic original or relevant in the field? Does it
address a specific gap in the field? This topic is not new in this field. A similar study was previously published.
3. What does it add to the subject area compared with other published
material? Not much addition to the subject. 
4. What specific improvements should the authors consider regarding the
methodology? What further controls should be considered? This study lacks detailed analysis.
5. Are the conclusions consistent with the evidence and arguments presented
and do they address the main question posed? Not applicable.
6. Are the references appropriate? Okay
7. Please include any additional comments on the tables and figures. The Figure's outline is not well defined.

Author Response

We would like to thank our reviewer for his comments.

Here is our point-by-point reply to his comments:

  1. I strongly suggest that authors learn the published article on this topic, Int. J. Environ. Res. Public Health 2022, 19(22), which is more comprehensive than the current manuscript.  This content of current manuscript is not enough.

Reply: We reviewed the suggested article and we found significant differences between the current manuscript and this article despite discussing the cardiovascular risk according to age and gender. however, we added the article to our references list as it was conducted in Saudi Arabia as well.

In the current study, we focused mainly on the relationship between age and the reproductive profile of women and their influence on cardiovascular risks. we recruited more than 1900 women compared to the 200 in the mentioned article. we assessed smoking, dietary habits, and physical activities among all participants in addition to all laboratory tests related to cardiovascular risks. We also measured cardiovascular risk scores with the Framingham score which is validated across different populations from all over the world.

2. This topic is not new in this field. A similar study was previously published. 

Reply: As we know that cardiac risk is a very important research topic and many studies had been published on this topic, to our knowledge, most of the studies which investigated cardiovascular diseases and risk factors in women in Saudi Arabia were either small cross-sectional studies targeting young college students, mostly investigating one risk factor for CVD, or studies which included both men and women but did not include analysis to investigate the influence of age, or the hormonal changes of pregnancy and the menopause on the CVD risk factors.

 3. Not much addition to the subject. 

Reply: As we clarified, we believe this is the first study with this large number of Saudi women with completed sociodemographic, clinical, and laboratory data to be available in the literature.

4. This study lacks detailed analysis

Reply: we conducted all the required analyses to reach the answers to our hypothesis and we definitely welcome any other suggested analysis to clarify any further inquiry.

5. The Figure's outline is not well defined.

Reply: We apologize for this unintended mistake that happened while uploading the manuscript and we already corrected the figures and tables to be clearly visible.

Reviewer 3 Report

The research paper entitled "Effects of age, metabolic and socioeconomic factors on cardiovascular risk among Saudi women: A subgroup analysis from the Heart Health Promotion (HHP) Study" is interested and have significant content. 

Following minor points should be considered.

1. The figures and tables are not properly visible. Please re-incorporate it in proper formatting.

2. Inter-age-group analysis is required to find the differences between the groups in each age-group. 

3. Review the typographical mistakes.

Author Response

We would like to thank our reviewer for his clear comments.

Kindly find our point-by-point reply to his suggestions:

  1. The figures and tables are not properly visible. Please re-incorporate it in proper formatting.

Reply: we would like to apologize for this unintended mistake that occurred while uploading the manuscript, and we corrected all figures and tables.

2. Inter-age-group analysis is required to find the differences between the groups in each age-group. 

Reply: As tables were not uploaded correctly, the statistical tests and the p-values of all comparisons were not properly shown. After corrections, all statistical analyses of all age groups are inserted in the tables and clearly visible.

3. Review the typographical mistakes.

Reply: Thorough language review was done to the manuscript and all revisions are highlighted.

Round 2

Reviewer 1 Report

The authors correctly answered the questions posed and significantly improved the quality of the manuscript.

Some minor revisions remain to be made:

1. Use the journal's template

2. Checking the correctness of titles and captions for tables and figures

3. The authors should include in the study limitations the fact that they did not use SCORE2 & SCORE2-OP as recommended by the European Society of Cardiology guidelines published in 2021 (European Heart Journal. 2021; 42:3227-3337 and freely available at https://www.escardio.org/Guidelines/Clinical-Practice-Guidelines/2021-ESC-Guidelines-on-cardiovascular-disease-prevention-in-clinical-practice). Therefore, they should also argue why it was not used and in any case I would cite these guidelines in the bibliography.

In conclusion, I strongly believe that this manuscript is very valuable because it deals appropriately with a topic (cardiovascular diseases in women) that still needs to be explored also in other countries. 

Author Response

Dear Reviewer, 

Thank you so much for your valuable comments.

We are sorry for not using the journal template, the tables were very large to fit in easily and as you know it is optional to use the template, and I hope you find the PDF version clear.

we revised the legends and captions of tables and figures.

We added to the limitation section that we did not used the SCORE2 and SCORE2-OP and referred to its reference. We are planning to re-analyze our dataset using these scores for future research report.

All our best regards

Reviewer 2 Report

I am fine with authors' reply. No more comment is needed from my side.

Author Response

Thank you, our reviewer, for your time and effort.